# New Flexible Tactile Sensor Based on Electrical Impedance Tomography

**DOI:** 10.3390/mi13020185

**Published:** 2022-01-26

**Authors:** Haibin Wu, Bingying Zheng, Haomiao Wang, Jinhua Ye

**Affiliations:** School of Mechanical Engineering and Automation, Fuzhou University, Fuzhou 350116, China; wuhb@fzu.edu.cn (H.W.); zhengbying@gmail.com (B.Z.); wanghaomiao.cn@gmail.com (H.W.)

**Keywords:** human–robot interaction, tactile sensor, position and force detection, electrical impedance tomography, conductivity

## Abstract

In order to obtain external information and ensure the security of human–computer interaction, a double sensitive layer structured tactile sensor was proposed in this paper. Based on the EIT (Electrical Impedance Tomography) method, the sensor converts the information from external collisions or contact into local conductivity changes, and realizes the detection of one or more contact points. These changes can be processed into an image containing positional and force information. The experiments were conducted on the actual sensor sample. The OpenCV toolkit was used to process the positional information of contact points. The distributional regularities of errors in positional detection were analyzed, and the accuracy of the positional detection was evaluated. The effectiveness, sensitivity, and contact area of the force detection were analyzed based on the result of the EIT calculations. Furthermore, multi-object tests of pressure were conducted. The results of the experiment indicated that the proposed sensor performed well in detecting the position and force of contact. It is suitable for human–robot interaction.

## 1. Introduction

With the increase in intelligence and the growth of security demands, robots need to interact more with the external environment, such as collaborative robots, mobile robots, and medical robots. The basic requirement of that is to be able to obtain external information. Tactile generally refers to human access to external information through the skin [1]. The tactile sensor is focused on the external informational acquisition for the robot and has received wide attention in human–robot interactions [2]. According to their functional principle, they can be divided into resistive tactile sensors [3], capacitive tactile sensors [4], piezoresistive tactile sensors [5], piezoelectric tactile sensors [6], optical tactile sensors [7], magnetic tactile sensors [8], and ultrasonic tactile sensors [9]. According to their structure, they can also be divided into array sensors and non-array sensors [10]. With the rise of MEMS, the array sensor has developed rapidly, and has the form of piezoresistive, piezoelectric, capacitive, etc., as mentioned above. The array sensor has the merits of high sensitivity, good dynamics, and high precision, which makes it suitable for the sensing of robot fingers. However, its structure is complex, with many wires, and blind areas of detection existent. Compared to the array sensor, the detection area of the non-array one is continuous, the signal processing circuit is simple and flexible. Non-array tactile sensors can also adapt to the surface by cutting some materials without damaging their own sensing function [11], they are easier to achieve large coverage and they are suitable for wrapping on large surfaces similar to robot arms, legs, and so on.

EIT is a process of estimating the internally and spatially varying electrical conductivity distribution in a region based on the observed current–voltage relationship at the boundary [12]. The tactile sensor based on EIT is one of the non-array structures [13,14]. It is characterized by the fact that the electrodes of the sensor are arranged on the boundary of the sensing material, it has good flexibility, and even has stretchability [15] (depending on the sensing material). Many researchers have researched the EIT-based tactile sensor. The sensor proposed by Kato et al. [16] can detect the single point and multi-point contact, and image for the pressure distribution. Nagakubo et al. [17] proposed a tactile sensor, and it can detect sophisticated external stimulation, including rubbing and pinching. Compared to the two kinds of stimulation in the reference [17], Silvera et al. [18] presented a tactile sensor which can identify eight different contact patterns using the LogitBoost algorithm. With 71% of accuracy, it is close to that of human recognition. The tactile sensor proposed by Visentin et al. [19] can not only detect the contact position, but also shows the potential to identify wrapped objects of different shapes. Yoon et al. [20] proposed a tactile sensor, which can detect the continuous movement of the contact in real-time. In the other literature of Visentin et al. [21], the body of the soft robot was utilized as the sensor, and the elongation and bending can be detected using EIT.

Efforts have been made to improve the quality of reconstructed images from EIT and to apply them to, for example, the recognition of force patterns or the recognition of object shapes. However, limited by the detection principle, the EIT-based tactile sensor has low detection accuracy, which leads to challenges in positional detection. The existing research mainly presented the imaging of the EIT, while the accuracy of the positional detection and force measurement was not intensively analyzed.

In this paper, a new EIT-based flexible tactile sensor with higher detection accuracy was presented, which makes positional detection possible. The error in positional detection of the sensor was analyzed quantificationally, and the relation between the external forces and the results of EIT was discussed. The sensor transduced the information of collisions into the changes of the conductivity and then such changes were reconstructed. The OpenCV toolkit was used to analyze the reconstructed images so that it could output the coordinates of the detected contact points continuously. The proposed tactile sensor can also reflect the force amplitude and make multi-point detection. The sensor is flexible and can be wrapped on the robot conveniently.

## 2. Materials and Methods

### 2.1. EIT-Based Tactile Sensor

It is easily imaginable that a tactile sensor can be made by applying EIT to conductive materials. Some electrodes are set on the boundary of the conductive material. Those electrodes are used for injecting electrical currents and measuring potential distributions on the boundary, taking 16 electrodes as an example shown in Figure 1a. Each time two adjacent electrodes (called driving electrodes) are selected to inject a current and the voltage difference between two adjacent electrode pairs (not adjacent to the driving electrode) is measured and collected into a set of 13 measurements, as the potential values at the boundary of the current injection mode. Considering all 16 current injection modes, namely (1,2), (2,3), ..., (15,16), and (16,1), the obtained data contain 208 (16 × 13) measurements. It should be noted that the potential values at the boundary will be different even under the same current injection mode when the resistance distribution in the conductive material changes. So, each of the 16 current injection modes will produce a unique distribution of potentials at the boundary and they will have enough data to solve the internal resistance distribution. In other words, resistance distribution over the conductive material can be reconstructed. If the material conductivity can be changed, the variation will also be able to be reconstructed by comparison of two tomographies between consecutive electrical impedance data. For example, the conductivity of piezoresistive film which is often used for tactile sensors will be changed when it is pressed. This kind of dynamic imaging method is able to eliminate noise in the process of data processing, and can improve the accuracy of the image process. Furthermore, the time of calculation is short so that it can be used for real-time imaging and is suitable for tactile sensors.

However, up to now, there are very few quantitative measurements about EIT-based tactile sensors. For example, it is hard to get the accurate output of the coordinate of the position and force amplitude when the tactile sensor is pressed. One of the main reasons is that the piezoresistive film has different impedance variation characteristics in a horizontal and vertical direction when it is pressed, as shown in Figure 1b [22,23]. It can be seen that the change in impedance is very significant in the vertical direction, but very slight in the horizontal direction because the pressed force is vertical. Whereas the injected current from the driving electrodes is spread horizontally over the whole plane of the tactile sensor. The vertical press has a smaller effect on the spread of the horizontal current so that the change in potential at the boundary of the tactile sensor is unobvious. It leads to poor imaging accuracy for the pressed position detection, and it is nearly unidentifiable or even miscalculated especially when the pressed area is less, or the pressed force is slighter. At present, the EIT-based tactile sensors are normally used for the imaging of the pressed position, but not for the pressed force detection because of the poor sensitivity.

In order to improve the sensitivity and accuracy of the dynamic EIT imaging, a new structure including two piezoresistive sheets with the same resistance characteristics was proposed, as shown in Figure 1c. The two piezoresistive sheets are stacked together to form a double-layer structure. The 16 electrodes are evenly spaced around the lower layer and isolated from the upper one. The contact surface of two piezoresistive sheets is not very smooth in fact, and there are lots of micro-structures [24]. When there is no external force exerting on the sensor as shown in Figure 1d, the flow of the current in the lower layer is nearly unaffected by the upper piezoresistive sheet because the micro-structures at the contact surface of both the upper layer and lower layer cause less real contact area, and thus the upper layer has nearly no effect on horizontal flow current spread. However, the resistance in the contact area will decrease more obviously under the action of external force, as shown in Figure 1e. In this case, the more electric conductive path of horizontal direction is formed in the contact area because of the bigger deformation of the micro-structures. The horizontal spread flow current will be more concentrated when going through the low-impedance contact area between the upper and lower layer. The impedance in the contact area changes greatly, and the local electric field also changes more obviously. In this way, the sensitivity of the sensor to orthogonal pressure is improved. The comparison of flow current distribution between the single layer and double layers under the same pressed force is shown in Figure 2a,b. It can be seen that the flow current trajectory changes more obviously than the single layer. This will produce a relatively larger potential variation at the sensor boundary.

### 2.2. The Material and Fabricating Method for the Sensor

The conductivity/resistivity of sensitive material should be changed when an external force is applied. The resister-sensitive material can often be produced by some conductive particle mixed into a kind of elastic insulating substrate. The sensitive material used in this paper was Velostat. Velostat from Adafruit Industries is a pressure-sensitive conductive film made from polyolefin and nano carbon black, the film has a 0.1 mm thickness, its volume resistivity is no more than 500 Ω cm, and its surface resistivity is no more than 31 KΩ. It has an operating temperature range of −45 °C to 65 °C, and its physical properties are not dependent on humidity. The surface morphology of the piezoresistive film used in this paper was observed by performing field-emission scanning electron microscopy (FESEM) analysis as in Figure 3. The FE-SEM images show that the surface of the piezoresistive sheet is not very smooth in fact, and there are lots of micro-structures.

The electrodes were made by coating the conductive silver paste around the sensitive material. After curing, the conductive silver paste used in this paper had a square resistance of less than 15 mΩ and a hardness of more than 2 H. The adhesion of the paste was tested with 3M600 tape, and no peeling occurred.

The piezoresistive film was cut into a circle with a diameter of 100 mm and fixed on a PCB with 16 electrodes evenly distributed along the circumference, which was specially designed for the EIT tests. We applied silver paste on the piezoresistive film, connected it to the electrodes on the PCB, and waited for the silver paste to cure. Since the conductivity of a piezoresistive film changes significantly in the direction of pressure and less in the horizontal direction when pressed, this means that it is difficult for the EIT system to detect conductivity changes when the sensor is pressed in a direction perpendicular to the surface of the sensor, as the current in the sensor flows in a horizontal direction. We therefore used a two-layer structure. The cured silver paste electrodes were covered with insulating material and a 100 mm diameter circular piezoresistive film was cut to cover the underlying piezoresistive film. The double layers structure of the sensor sample using the Velostat piezoresistive film is shown in Figure 4. The contact impedance between the two piezoresistive films is large when the sensor is not subjected to external forces, and when the sensor is subjected to external forces, the contact impedance between the pressed areas decreases due to the effect of the press. This translates the effect of the vertically acting force to the horizontal direction.

### 2.3. Signal Process for EIT-Based Double Sheets Tactile Sensor

In biomedicine or industrial detection, the impedance of detective objects includes not only the resistive component (real part of impedance), but also the capacitive component (imaginary part of impedance). For obtaining the whole impedance, the AC signal with a certain frequency and amplitude is applied in the common EIT system. The electrode needs to switch frequently between being connected or disconnected from the object in above-mentioned EIT system, and the AC signal can effectually eliminate the effect of contact resistance caused by this process. However, what is measured of the circuit is also an AC signal, which includes a real part and imaginary part, and needs to be demodulated by phase-sensitive demodulation technology. The sensitive component selected in this paper is a resistive material whose resistance component can reflect the entire impedance and thus does not require frequent switching of the electrodes. Moreover, the sampling frequency of the AC signal is much higher than that of the DC signal, and the conversion rate of A/D is also higher. In a word, using the DC signal is convenient. It can simplify the circuit.

A simple circuit system was designed for sensor data acquisition, as shown in Figure 5. The positive and negative electrodes of the constant current source are connected to different multi-channel switches, which are controlled by a single-chip microcomputer and further connected to the driving electrodes of sensitive materials. Similarly, the measuring electrode that is connected to the acquisition circuit at a certain time is also controlled by the single-chip computer. The voltage difference between the measuring electrode and the negative electrode of the constant current source is taken as the measured voltage. After the signal is amplified, the data acquisition card USB-6210 of NI is used to collect and transmit it to the computer. The computer sends a square wave signal to the acquisition card every certain time and transmits it to the single-chip microcomputer through the acquisition card. Adopting adjacent driving mode, the system switches the multiplex switch to the adjacent electrode for measurement when the level of square wave signal changes was detected by a single-chip microcomputer. The voltage differences saved by the computer are the measured voltage, which is then processed into the voltage differences between adjacent electrodes. The whole actual system of the proposed tactile sensor is shown in Figure 6.

Currently, there are many reconstructing algorithms, including the back-projection algorithm, Newton-Raphson method, singular value decomposition, etc., which can be divided into static imaging and dynamic imaging. The dynamic imaging method is able to eliminate noise in the process of data processing and has better accuracy. The back-projection algorithm was used in this dynamic imaging method. The relative distribution image of resistance in the conductor is reconstructed by using the EIT toolkit—pyEIT using Python proposed by Liu et al. [25].

Supposing that a piece of sensitive sheet is wrapped on the robot, the conductivity of the material changes when the external force is applied. From the current injected and the voltage measured from the boundary, respectively, the relative distribution image of resistance in the conductor can be reconstructed. The area where the resistivity changes is the position where the external force is applied, and the magnitude of the change reflects the magnitude of the applied force. In this way, the sensor successfully realizes the detection of the external force acting on the robot.

## 3. Experiments

To demonstrate the performance of the proposed sensor, a series of experiments were designed to test the system and the results were analyzed. Firstly, the measurements of the double-layer structure sensor in imaging were carried out contrasting to the single layer. Secondly, the detection of the contact position was achieved using the OpenCV toolkit and the precision was assessed. Then, the effect of pressure detection was analyzed. Finally, tests for multi-object detection were carried out.

### 3.1. The Demonstration of Imaging

The reconstructed image of the single-layer structure sensor under a force of 60 N with the contact area of 226.98 mm^2^ was shown in Figure 7a, where the force was applied in the center of the detection area, the reconstructed image did not show the true position of contact. Figure 7b showed the imaging of the double-layer structure sensor with the same contact area, even with an applied force of only 0.5 N, the sensor can sensitively detect the contact. The results showed that the double-layer structure sensor has improved the sensitivity greatly.

### 3.2. The Positional Detection Using OpenCV

The OpenCV toolkit was used to find the centroid of the loaded area based on the reconstructed image. Figure 8 showed the flowchart of image processing.

A reconstructed image is needed firstly, and the image processing procedure includes: extracting the RIO (region of interest), gray processing, adaptive threshold binarization algorithm, and open operation to eliminate the noises and discontinuities. The centroid coordinates of the loaded area were gained using the detected edge. At last, the coordinates of the image were converted to the coordinates of the sensor.

In order to get to know the accuracy of positional measurement in different areas over the whole sensor, as shown in Figure 9, a 100 g weight was placed on the double-layer structure sensor, and the weight was moved by a step of 5 mm with the direction of horizontal, vertical, and 45°, respectively. The actual coordinates of the weight are compared with the centroid coordinates calculated by OpenCV demonstrated above.

The error in positional detection was defined as the Euclidean distance between the actual coordinates of weight and the centroid coordinates of the reconstructed image detected by OpenCV. Figure 10 showed the diagram of the detected error. The errors were symmetrically distributed about the center of the detective region. The error achieved a minimum around the center area and increased with the distance from the center. The reason was inferred as follows. The reconstructed image was a composition of all 16 back-projection images under different driving electrodes. Figure 11a,b showed that the back-projection was projected to the middle region evenly, but to the fringe region unevenly. The EIT imaging algorithm is more sensitive to locations close to the electrodes than to locations far from the electrodes where small noise can lead to greater errors. The maximum absolute errors shown in Figure 10 do not exceed 11 mm, all within the circular domain of the loaded weights, which is the circular domain with a radius of 11.5 mm, indicating that the accuracy of the positional detection of the sensor described in this paper can meet the requirements of human–machine interaction.

### 3.3. Pressure Detection

#### 3.3.1. Single Point Test

Different pixel values represent different degrees of conductivity change in a reconstructed image. A bigger load leads to a more obvious change in conductivity, which means a higher pixel value. Thus, a corresponding relationship between the magnitude of external pressure and the relative change in conductivity can be founded. Figure 12 shows three different weights stacked together, which ensure the contact area remains unchanged. Under a fixed contact area, weights from 100 g to 3900 g were loaded over the sensor. Figure 13 shows the curve demonstrating the relationship between the magnitude of weights and the relative change in conductivity in the reconstructed image. In general, the value increased with the mass of the weight, and it is an approximate linear relationship during a certain range, this shows that the EIT method can be used to identify the relative magnitude of the pressure. As weight increases, the curve tends to be flat, which means the change of conductivity was close to its limit.

#### 3.3.2. Sensitivity Analysis

In order to get to know the sensitivity of pressure detection in different areas over the whole sensor, the same weight was moved in three directions similar to the condition demonstrated in Figure 9. The relative change in conductivity with the weight in different places was shown in Figure 14.

The relative change in conductivity has a more significant change in the fringe region than in the center region. This is probably because the conductivity change around electrodes caused a more significant change of measured potential. That is to say, the sensitivity of the fringe region is higher than the center region under the same pressure.

Figure 15 showed a 100 g weight on the sensor. In this test, the weight is placed on 16 points evenly distributed along a circle with a radius of 30 mm.

Figure 16 showed the result of a different measurement point. It can be seen that the relative change in conductivity of different points around a circle was at roughly the same level. This indicates that the same relative change in conductivity at the same distance from the center of the detection region can indicate the same pressure. The pole diameter of the press position has an effect on the relative change in conductivity, and the larger the pole diameter, the greater the rate of change, while the pole angle has almost no effect on it. Therefore, the distance between the target position and the center of the detection area should be considered when the relative change in conductivity is used to express the force magnitude.

#### 3.3.3. Test under Different Contact Areas

The same quality under different contact areas may lead to different pixel values in reconstructed images. In this test, contact areas were designed to a circular region with areas of 415.5 mm^2^, 615.8 mm^2^ and 1134.1 mm^2^, and a total mass of the weights was 800 g for all three tests shown in Figure 17a–c. Theoretically, on the one hand, in the case where the pressure is constant, the intensity of pressure decreased with the increase of contact area, and there should be a downward trend in the pixel value. On the other hand, with the increase of contact area, there are more areas in which the conductivity changes, so the value of pixel value should also tend to increase. Figure 17d–f were corresponding reconstructed images. The maximums of pixel values were 11.12, 15.02, and 29.8, respectively. It can be seen that the maximum pixel value increased with the area of contact. That is, the influence of the contact area was larger than that of the intensity of pressure in fact.

#### 3.3.4. Multi-Object Test

Figure 18a showed two 100 g weights placed on specified points belonging to two different circles with a radius of 20 mm and 35 mm, respectively. Figure 18b showed the reconstructed images. It can be seen that the closer the contact point was to the edge, the more obvious the change of conductivity was.

Figure 19a showed a 100 g weight and a 200 g weight placed on two opposing points belonging to the same circle with a radius of 25 mm. Figure 19b showed the reconstructed image. The image indicated that the pixel value of 100 g weight was lower than that of 200 g weight. Hence, the proposed sensor can roughly recognize different pressures under multi-object detection.

## 4. Conclusions

This paper proposed an EIT-based double piezoresistive sheet structured tactile sensor. The applicability of EIT in the tactile sensor was elaborated. The sensor is appropriate for sensing the position of pressure over the whole sensitive area, can roughly estimate the pressure amplitude, and can detect multi-point contact. A sensor with a circular structure and 16 electrodes was designed. The DC signal was selected to inject into the sensor. The positional detection using the OpenCV toolkit was innovatively proposed based on the image of EIT. The distribution and accuracy of positional detection were analyzed according to the result of the detection. The result showed that the error of positional detection increased along with the direction of radius. In other words, the accuracy was lower when close to the boundary of the detective region. The estimate for pressed force was also analyzed based on the results of EIT. The results showed that the pixel value of the pressed area approximately linearly relates to the loaded force under the same contact position and the certain range of pressure. The sensitivity of force detection gradually increases from the central area to the boundary of the sensor. It was also influenced by the contact area. In multi-point contact detection, different contact forces can be distinguished by the sensor under the one-factor-at-a-time method. It indicates that the proposed sensor is proper for the robot to get more information about the external contacts.

## Figures and Tables

**Figure 1 micromachines-13-00185-f001:**
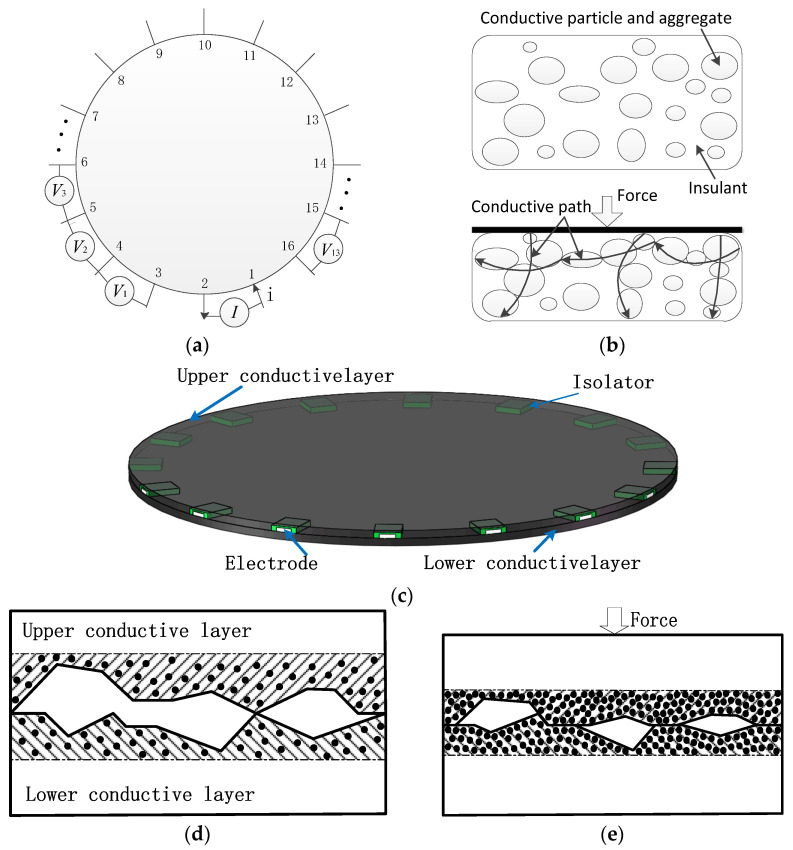
The structure and principle of double-layer piezoresistive sheets tactile sensor: (**a**) The schematic of sensor; (**b**) Microcosmic mechanism after the piezoresistive film is pressed; (**c**) Three-dimensional model of the sensor with two-layer structure; (**d**) Schematic diagram of high impedance state; (**e**) Schematic diagram of low impedance state.

**Figure 2 micromachines-13-00185-f002:**
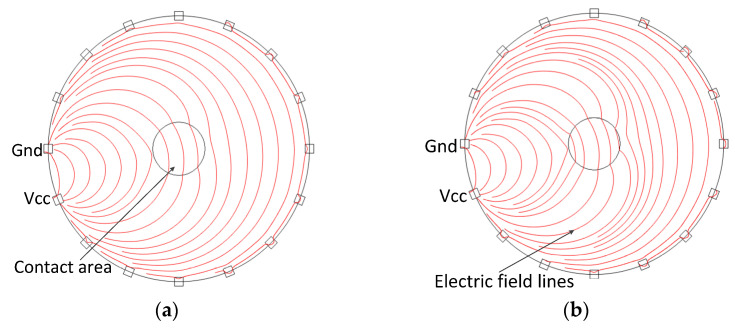
Current density comparison: (**a**) single-layer structure; (**b**) two-layer structure.

**Figure 3 micromachines-13-00185-f003:**
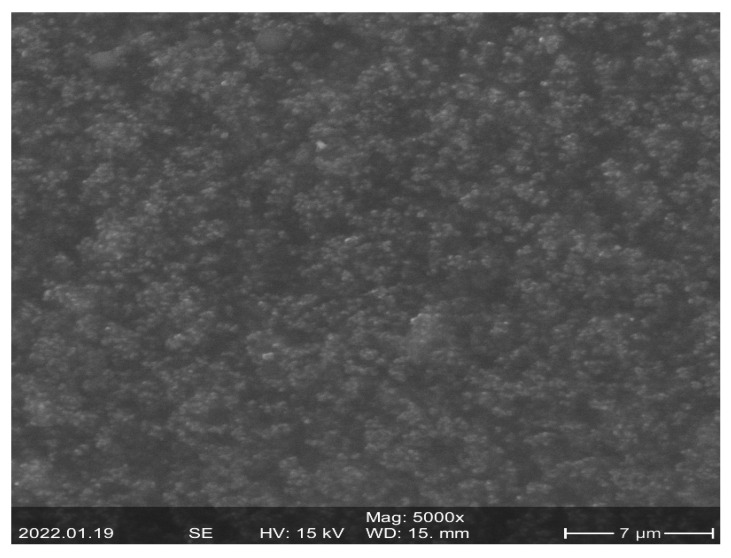
The surface morphology of the piezoresistive film.

**Figure 4 micromachines-13-00185-f004:**
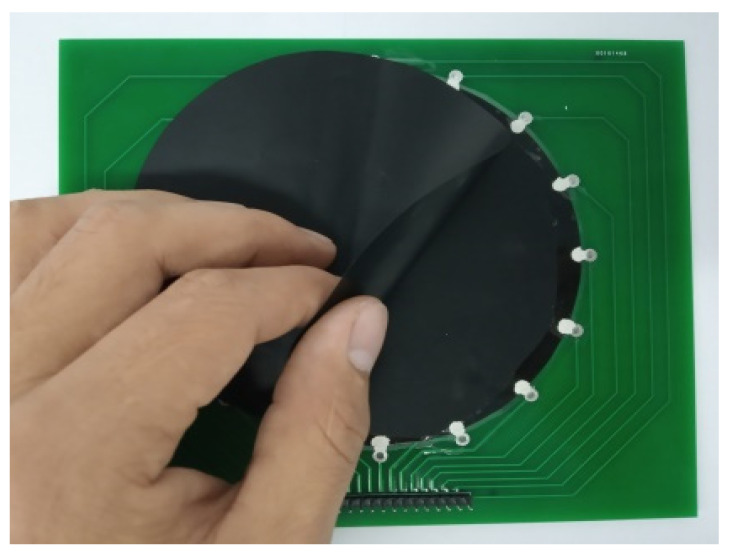
Two-layer structural sensor.

**Figure 5 micromachines-13-00185-f005:**
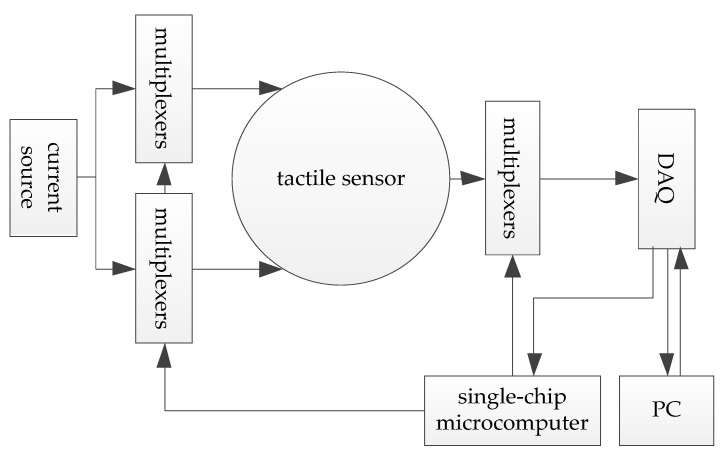
The diagram of the circuit.

**Figure 6 micromachines-13-00185-f006:**
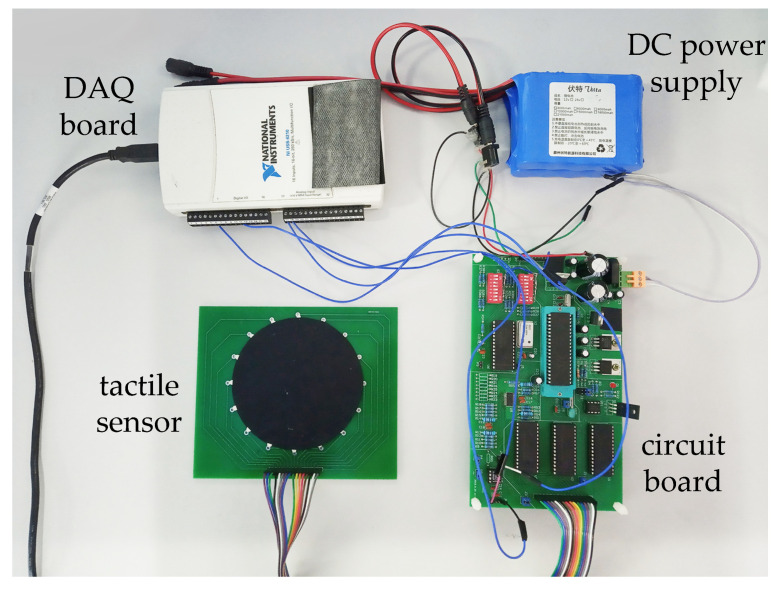
The hardware system of proposed sensor.

**Figure 7 micromachines-13-00185-f007:**
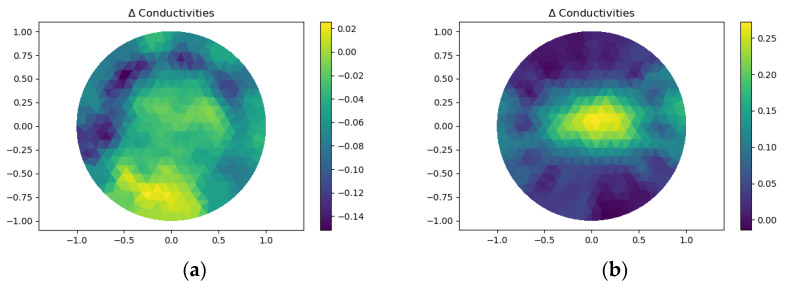
(**a**) Reconstructed image of single-layer structure sensor; (**b**) Reconstructed image of double-layer structure sensor.

**Figure 8 micromachines-13-00185-f008:**
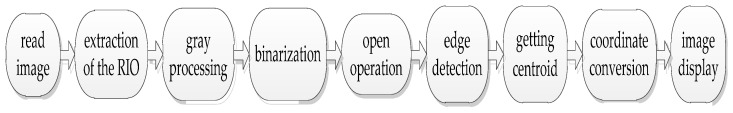
The flowchart of the image processing.

**Figure 9 micromachines-13-00185-f009:**
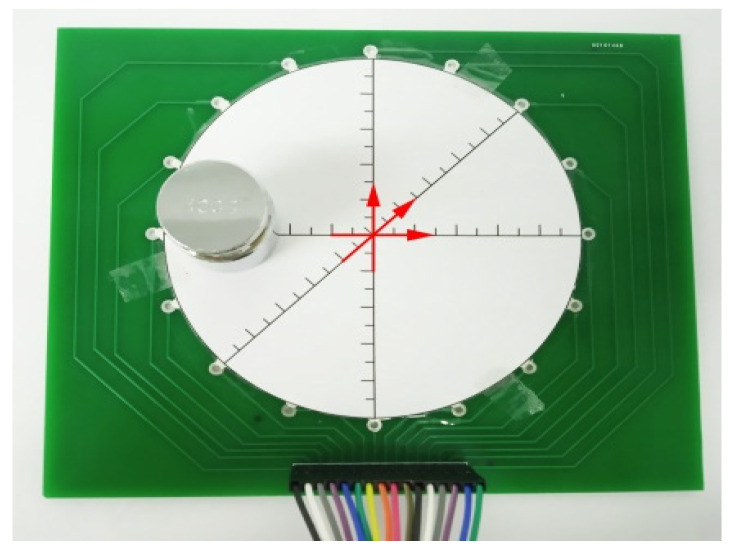
The device of positional detection in OpenCV.

**Figure 10 micromachines-13-00185-f010:**
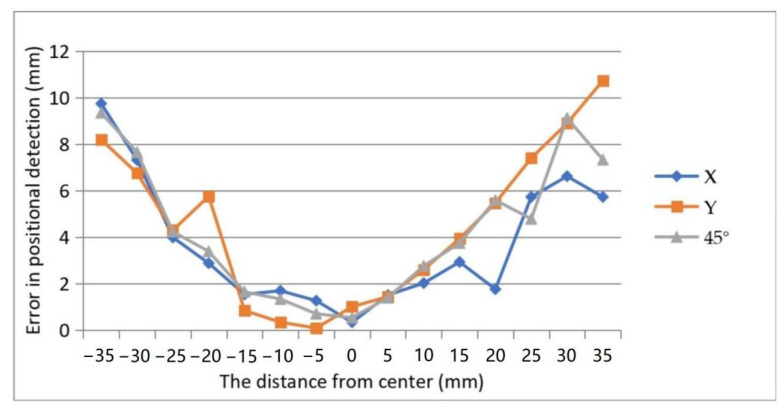
Errors in positional detection by experiment.

**Figure 11 micromachines-13-00185-f011:**
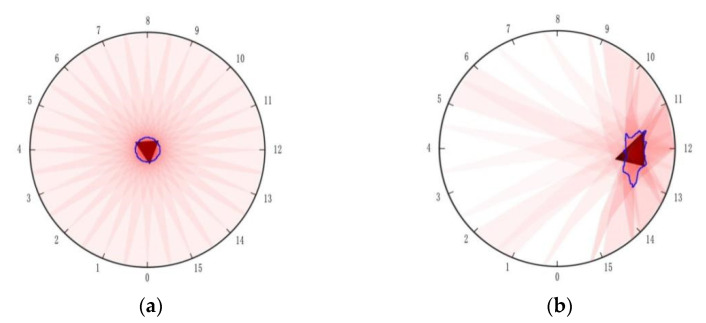
(**a**) The object lying in the central region; (**b**) The object lying in the fringe region.

**Figure 12 micromachines-13-00185-f012:**
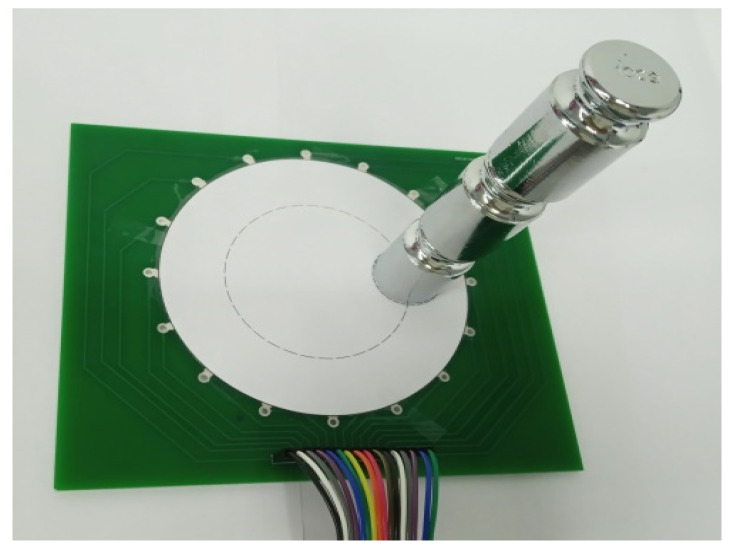
Setup to detect the relationship between pressure and the value color bar.

**Figure 13 micromachines-13-00185-f013:**
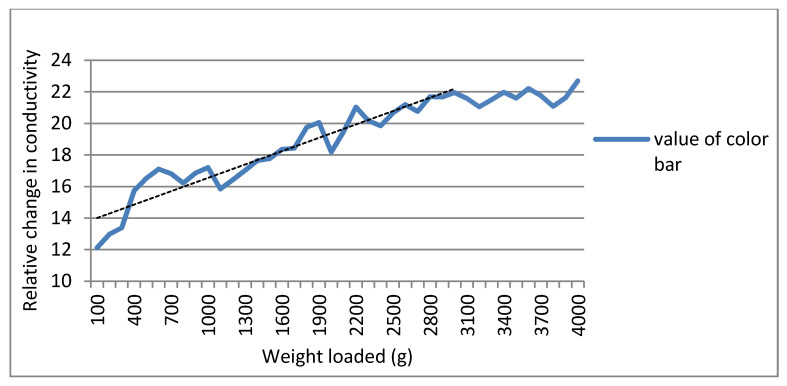
Relationship between the magnitude of weights and the relative change in conductivity.

**Figure 14 micromachines-13-00185-f014:**
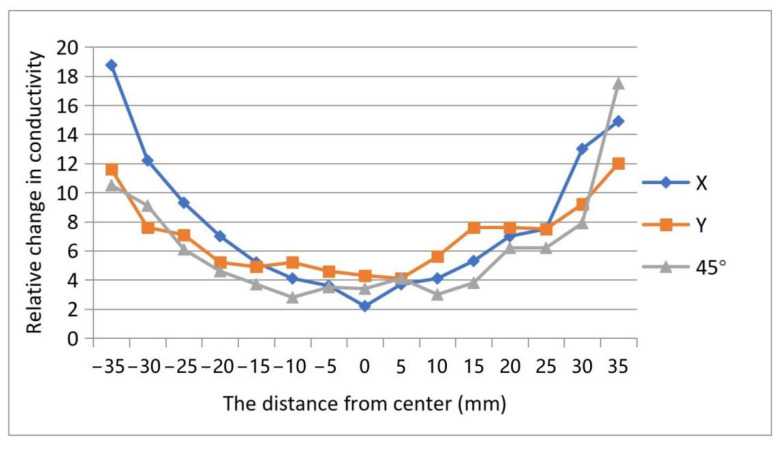
The sensitivity in linear direction.

**Figure 15 micromachines-13-00185-f015:**
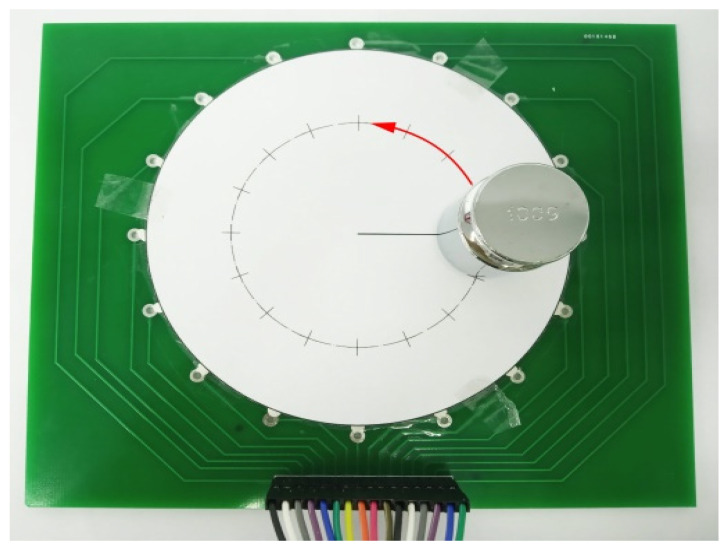
Setup to detect the sensitivity in circular direction.

**Figure 16 micromachines-13-00185-f016:**
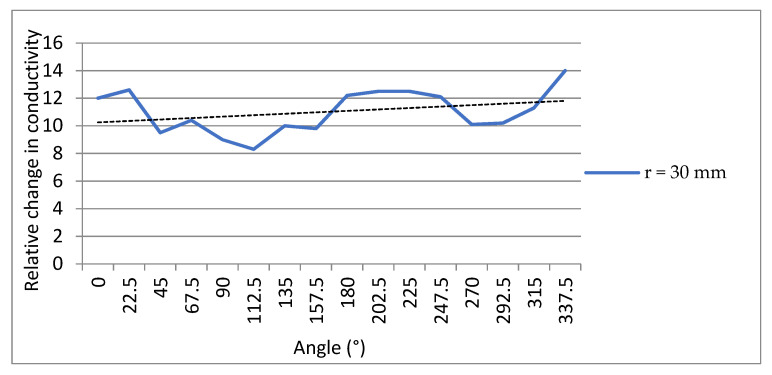
The sensitivity in circular direction.

**Figure 17 micromachines-13-00185-f017:**
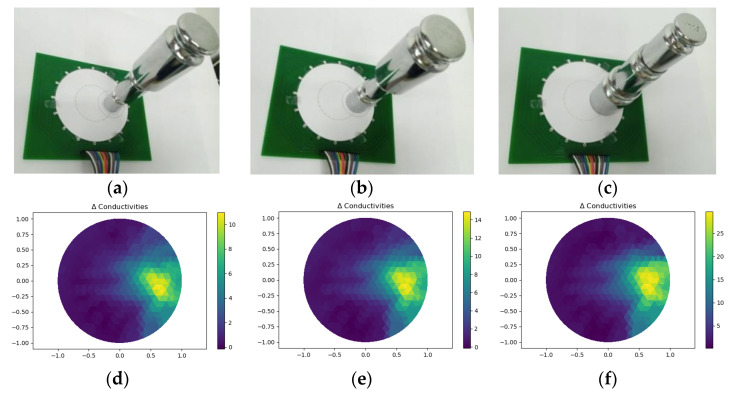
(**a**~**c**) Setup under different contact areas; (**d**~**f**) Reconstructed images.

**Figure 18 micromachines-13-00185-f018:**
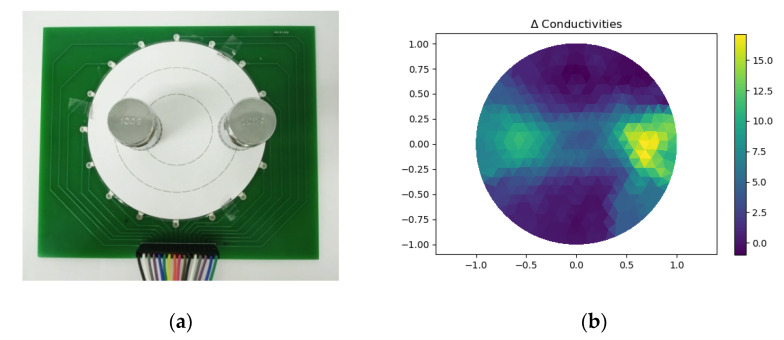
(**a**) Setup to test multi-object which were placed on different circles; (**b**) Reconstructed image.

**Figure 19 micromachines-13-00185-f019:**
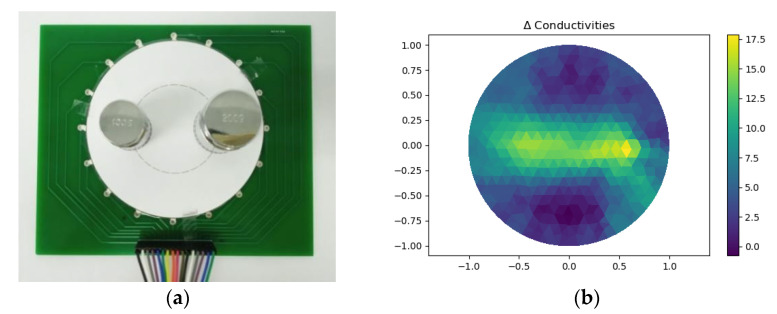
Different pressure loading under multi-object: (**a**) Setup of experiment; (**b**) Reconstructed image.

## Data Availability

No new data were created or analyzed in this study. Data sharing is not applicable to this article.

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
