# Peer review of "New Flexible Tactile Sensor Based on Electrical Impedance Tomography"

_micromachines, 2022, doi:10.3390/mi13020185_

Round 1
Reviewer 1 Report
This manuscript reports a double sensitive layer structured tactile sensor considering the electrical impedance tomography (EIT) method. Thus, this sensor can convert the collisions or contact into local conductivity variations and obtain the detection of one or more points. In addition, these variations are processed into an image that can contain the position and force information. The experimental results showed that the reported sensor has a good performance to detect the position and force of contact. This sensor could be used for applications with human-robot interaction. This manuscript could be improved considering the following comments:
1.-This manuscript should add more recent references (e.g., between 2019 and 2021).
2.- Introduction section could include the main advantages of the proposed sensor in comparison with other tactile sensors reported in the literature.
3.-More technical data about different parameters of the materials used in the fabrication process of the sensor could be added.
4.-Which are the main limitations or challenges of the proposed sensor?
5.-What is the influence of the temperature and humidity in the sensor performance?
6.-Quality of the figure 7 should be improved.
7.-Authors could include more discussion about results reported in figures 9, 12, 13, and 15.
8.-What is the future research work?
Reviewer 2 Report
In the manuscript of micromachines-1558701, a tactile sensor with double sensitive layer structure is studied, which can detect contact points by converting collision or contact information into local conductivity change. The toolkit of OpenCV is used to analyze the position information of contact points and detect the error distribution. However, compared with the previous reference, the research is lack of novelty, so I suggest rejecting the manuscript and offer the following comments for consideration.
- In the section of introduction, the authors need to point out the innovation of the article further.
- The detailed information of the sensing film and electrodes should be provided, including the specific ratio of the compound material and the electrical conductivity level.
- The difference of single-layer and two-layer sensor should be mentioned in detail, including the roughness of the sensing films.
- The author should give the measurement accuracy of the tactile
- Could the authors give the experimental results to detect more than two points?
- Why did the authors use the toolkit of OpenCV?
- In the Table 1, less information of previous references is provided, so a detailed comparison table may be necessary.
Some typos:
- The text error at the top of figure 1.
- The font of the first sentence of the second paragraph of section 2.3 is wrong.
- The text in Figure 4 is not clear enough.
- The year in reference [18] was marked incorrectly.
- Check the quality of all figures and their layout problems.
Round 2
Reviewer 1 Report
Authors have improved their manuscript based on reviewer's comments.
Reviewer 2 Report
All the comments and suggestions are addressed, and I recommend the manuscript to be accepted.